# Neural Belief Tracker: Data-Driven Dialogue State Tracking

## Abstract

One of the core components of modern spoken dialogue systems is the *belief tracker*, which estimates the user's goal at every step of the dialogue. However, most current approaches have difficulty scaling to larger, more complex dialogue domains. This is due to their dependency on either: **a)** Spoken Language Understanding models that require large amounts of annotated training data; or **b)** hand-crafted lexicons for capturing some of the linguistic variation in users' language. We propose a novel Neural Belief Tracking (NBT) framework which overcomes these problems by building on recent advances in representation learning. NBT models reason over pre-trained word vectors, learning to compose them into distributed representations of user utterances and dialogue context. Our evaluation on two datasets shows that this approach surpasses past limitations, matching the performance of state-of-the-art models which rely on hand-crafted semantic lexicons and outperforming them when such lexicons are not provided.

## 1 Introduction

Spoken dialogue systems (SDS) allow users to interact with computer applications through conversation. Task-based dialogue systems help users achieve goals such as finding restaurants or booking flights. The *dialogue state tracking* (DST) component of an SDS serves to interpret user input and update the *belief state*, which is the system's internal representation of the state of the conversation (Young et al., 2010). This is a probability distribution over dialogue states used by the downstream dialogue manager component

to decide which action the system should perform next. The Dialogue State Tracking Challenge (DSTC) series of shared tasks has provided a common evaluation framework accompanied by labelled datasets (Williams et al., 2016). In this framework, the dialogue system is supported by a *domain ontology* which describes the range of user intents the system can process. The ontology defines a collection of *slots* and the *values* each slot can take. The system must track the search constraints expressed by users (*goals* or *informable* slots) and questions the users ask about search results (*requests*), taking into account each user utterance (input via a speech recogniser) and the dialogue context (e.g., what the system just said). The following example shows the true state after each user utterance in a three-turn conversation:

> **User:** I'm looking for a cheaper restaurant
> `inform(price=cheap)`
> **System:** How about Thai food?
> **User:** Yes please
> `inform(price=cheap, food=Thai)`
> **System:** The House serves cheap Thai food
> **User:** Where is it?
> `inform(price=cheap, food=Thai);`
> `request(address)`
> **System:** The House is at 106 Regent Street

DST models depend on identifying mentions of ontology items in user utterances, which becomes a non-trivial task when confronted with lexical variation, the dynamics of context and noisy speech recognition. Some approaches assume that a separate Spoken Language Understanding (SLU) module will solve this problem for them. However, such models require vast amounts of annotated training data. Moreover, coupling SLU and DST in a single model has been shown to improve belief tracking performance (Henderson et al., 2014d). On the other hand, such coupled models typically rely

**FOOD=CHEAP:** [affordable, budget, low-cost, low-priced, inexpensive, cheaper, economic, ...]

**RATING=HIGH:** [best, high-rated, highly rated, top-rated, cool, chic, popular, trendy, ...]

**AREA=CENTRE:** [center, downtown, central, city centre, midtown, town centre, ...]

Figure 1: An example semantic dictionary with rephrasings for three ontology values in a *restaurant search* domain.

on *manually constructed* semantic dictionaries to identify alternative mentions of ontology items that vary lexically or morphologically. Figure 1 gives an example of such a dictionary for three slot-value pairs. This approach, which we term *delexicalisation*, is clearly not scalable to larger, more complex dialogue domains.

In this paper, we present two new models, collectively called the Neural Belief Tracker (NBT) family. The proposed models couple SLU and DST, efficiently learning to handle variation without requiring *any* hand-crafted resources. To do that, NBT models move away from exact matching and instead reason entirely over pre-trained word vectors. The vectors making up the user utterance and preceding system output are first composed into intermediate representations. These are then used to decide which of the ontology-defined intents have been expressed by the user up to that point in the conversation.

To the best of our knowledge, NBT models are the first to successfully use pre-trained word vector spaces to improve the language understanding capability of belief tracking models. In evaluation on two datasets, we show that: **a)** NBT models match the performance of delexicalisation-based models which make use of hand-crafted semantic lexicons; and **b)** the NBT models significantly outperform those models when such resources are not available. Consequently, we believe this work proposes a framework better-suited to scaling belief tracking models for deployment in real-world dialogue systems operating over sophisticated application domains where the creation of such domain-specific lexicons would be infeasible.

## 2 Background

Models for probabilistic dialogue state tracking, or *belief tracking*, were introduced as components of spoken dialogue systems in order to better handle noisy speech recognition and other sources of uncertainty in understanding a user's goals (Bohus and Rudnicky, 2006; Williams and Young, 2007; Young et al., 2010). Modern dialogue management policies can learn to use a tracker's distribution over intents to decide whether to execute an action or request clarification from the user. As mentioned above, the DSTC shared tasks have spurred research on this problem and established a standard evaluation paradigm (Williams et al., 2013; Henderson et al., 2014b,a). In this setting, the task is defined by an *ontology* that enumerates the goals a user can specify and the attributes of entities that the user can request information about. Many different belief tracking models have been proposed in the literature, from generative (Thomson and Young, 2010) and discriminative (Henderson et al., 2014d) statistical models to rule-based systems (Wang and Lemon, 2013). To motivate the work presented here, we categorise prior research according to their reliance (or otherwise) on a separate SLU module for interpreting user utterances:[1]

**Separate SLU**: Traditional SDS pipelines use Spoken Language Understanding (SLU) decoders to detect slot-value pairs expressed in the Automatic Speech Recognition (ASR) output. The downstream DST model then combines this information with the past dialogue context to update the belief state (Thomson and Young, 2010; Wang and Lemon, 2013; Lee and Kim, 2016; Perez, 2016; Perez and Liu, 2017; Sun et al., 2016; Jang et al., 2016; Shi et al., 2016; Dernoncourt et al., 2016; Liu and Perez, 2017; Vodolán et al., 2017). In the DSTC challenges, some systems used the output of template-based matching systems such as Phoenix (Wang, 1994). However, more robust and accurate statistical SLU systems are available. Many discriminative approaches to spoken dialogue SLU train independent binary models that decide whether each slot-value pair was expressed in the user utterance. Given enough data, these models can learn which lexical features are good indicators for a given value and can capture elements of paraphrasing (Mairesse et al., 2009). This line of work later shifted focus to robust handling of rich ASR output (Henderson et al., 2012; Tur et al., 2013). SLU has also been treated as a sequence labelling problem, where each word in an utterance is labelled according to its role in the user's intent;

---

[1]The best-performing models in DSTC2 all used both raw ASR output and the output of SLU decoders (Williams, 2014; Williams et al., 2016). This does not mean that those models are immune to the drawbacks identified here for the two model categories; in fact, they share the drawbacks of both.

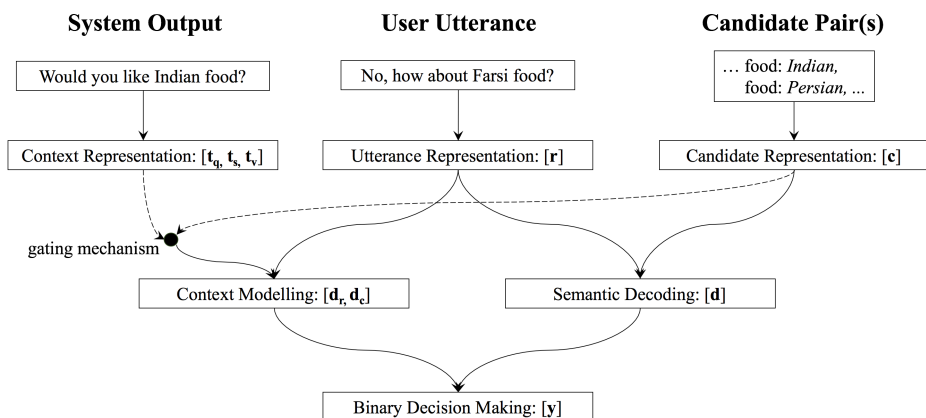

Figure 2: Architecture of the NBT Model. The implementation of the three representation learning subcomponents can be modified, as long as these produce adequate vector representations which the downstream model components can use to decide whether the current candidate slot-value pair was expressed in the user utterance (taking into account the preceding system act).

standard labelling models such as CRFs or Recurrent Neural Networks can then be used (Raymond and Ricardi, 2007; Yao et al., 2014; Celikyilmaz and Hakkani-Tur, 2015; Mesnil et al., 2015; Peng et al., 2015; Zhang and Wang, 2016; Liu and Lane, 2016b; Vu et al., 2016; Liu and Lane, 2016a, i.a.). Other approaches adopt a more complex modelling structure inspired by semantic parsing (Saleh et al., 2014; Vlachos and Clark, 2014). One drawback shared by these methods is their resource requirements, either because they need to learn independent parameters for each slot and value or because they need fine-grained manual annotation at the word level. This hinders scaling to larger, more realistic application domains.

**Joint SLU/DST**: Research on belief tracking has found it advantageous to reason about SLU and DST jointly, taking ASR predictions as input and generating belief states as output (Henderson et al., 2014d; Sun et al., 2014; Zilka and Jurcicek, 2015; Mrkšić et al., 2015). In DSTC2, systems which used no external SLU module outperformed all systems that only used external SLU features. Joint models typically rely on a strategy known as *delexicalisation* whereby slots and values mentioned in the text are replaced with generic labels. Once the dataset is transformed in this manner, one can extract a collection of template-like $n$-gram features such as **[want *tagged-value* food]**. To perform belief tracking, the shared model iterates over all slot-value pairs, extracting delexicalised feature vectors and making a separate binary decision regarding each pair. Delexicalisation introduces a hidden dependency that is rarely discussed: how do we identify slot/value mentions in text? For toy domains, one can manually construct *semantic*

*dictionaries* which list the potential rephrasings for all slot values. As shown by Mrkšić et al. (2016), the use of such dictionaries is essential for the performance of current delexicalisation-based models. Again though, this will not scale to the rich variety of user language or to general domains.

The primary motivation for the work presented in this paper is to overcome the limitations that affect previous belief tracking models. The NBT model efficiently learns from the available data by: **1)** leveraging semantic information from pre-trained word vectors to resolve lexical/morphological ambiguity; **2)** maximising the number of parameters shared across ontology values; and **3)** having the flexibility to learn domain-specific paraphrasings and other kinds of variation that make it infeasible to rely on exact matching and delexicalisation as a robust strategy.

## 3 Neural Belief Tracker

The Neural Belief Tracker (NBT) is a model designed to detect the slot-value pairs that make up the user's goal at a given turn during the flow of dialogue. Its input consists of the system dialogue acts preceding the user input, the user utterance itself, and a single candidate slot-value pair that it needs to make a decision about. For instance, the model might have to decide whether the goal FOOD=ITALIAN has been expressed in *'I'm looking for good pizza'*. To perform belief tracking, the NBT model *iterates* over all candidate slot-value pairs (defined by the ontology), and decides which ones have just been expressed by the user.

Figure 2 presents the flow of information in the model. The first layer in the NBT hierarchy performs representation learning given the three model

inputs, producing vector representations for the user utterance ($\mathbf{r}$), the current candidate slot-value pair ($\mathbf{c}$) and the system dialogue acts ($\mathbf{t_q}, \mathbf{t_s}, \mathbf{t_v}$). Subsequently, the learned vector representations interact through the *context modelling* and *semantic decoding* submodules to obtain the intermediate *interaction summary* vectors $\mathbf{d_r}$, $\mathbf{d_c}$ and $\mathbf{d}$. These are used as input to the final *decision-making* module which decides whether the user expressed the intent represented by the candidate slot-value pair.

### 3.1 Representation Learning

For any given user utterance, system act(s) and candidate slot-value pair, the representation learning submodules produce vector representations which act as input for the downstream components of the model. All representation learning subcomponents make use of pre-trained collections of word vectors. As shown by Mrkšić et al. (2016), specialising word vectors to express *semantic similarity* rather than *relatedness* is essential for improving belief tracking performance. For this reason, we use the semantically-specialised Paragram-SL999 word vectors (Wieting et al., 2015) throughout this work. The NBT training procedure keeps these vectors fixed: that way, at test time, unseen words semantically related to familiar slot values (i.e. *inexpensive* to *cheap*) will be recognised purely by their position in the original vector space (see also Rocktäschel et al. (2016)). This means that the NBT model parameters can be shared across all values of the given slot, or even across all slots.

Let $u$ represent a user utterance consisting of $k_u$ words $u_1, u_2, \ldots, u_{k_u}$. Each word has an associated word vector $\mathbf{u}_1, \ldots, \mathbf{u}_{k_u}$. We propose two model variants which differ in the method used to produce vector representations of $u$: NBT-DNN and NBT-CNN. Both act over the constituent $n$-grams of the utterance. Let $\mathbf{v}_i^n$ be the concatenation of the $n$ word vectors starting at index $i$, so that:

$$\mathbf{v}_i^n = \mathbf{u}_i \oplus \ldots \oplus \mathbf{u}_{i+n-1} \qquad (1)$$

where $\oplus$ denotes vector concatenation. The simpler of our two models, which we term NBT-DNN, is shown in Figure 3. This model computes cumulative $n$-gram representation vectors $\mathbf{r}_1$, $\mathbf{r}_2$ and $\mathbf{r}_3$, which are the $n$-gram 'summaries' of the unigrams, bigrams and trigrams in the user utterance:

$$\mathbf{r}_n = \sum_{i=1}^{k_u - n + 1} \mathbf{v}_i^n \qquad (2)$$

Each of these vectors is then non-linearly mapped to intermediate representations of the same size:

$$\mathbf{r}_n' = \sigma(W_n^s \mathbf{r}_n + b_n^s) \qquad (3)$$

where the weight matrices and bias terms map the cumulative $n$-grams to vectors of the same dimensionality and $\sigma$ denotes the sigmoid activation function. We maintain a separate set of parameters for each slot (indicated by superscript $s$). The three vectors are then summed to obtain a single representation for the user utterance:

$$\mathbf{r} = \mathbf{r}_1' + \mathbf{r}_2' + \mathbf{r}_3' \qquad (4)$$

The cumulative $n$-gram representations used by this model are just unweighted sums of all word vectors in the utterance. Ideally, the model should learn to recognise which parts of the utterance are more relevant for the subsequent classification task. For instance, it could learn to ignore verbs or stop words and pay more attention to adjectives and nouns which are more likely to express slot values.

Our second model, NBT-CNN, draws inspiration from successful applications of Convolutional Neural Networks (CNNs) for language understanding (Collobert et al., 2011; Kalchbrenner et al., 2014; Kim, 2014). These models typically apply a number of convolutional filters to $n$-grams in the input sentence, followed by non-linear activation functions and max-pooling. Following this approach, the NBT-CNN model applies $L = 300$ different filters for $n$-gram lengths of $1, 2$ and $3$ (Figure 4). Let $F_n^s \in R^{L \times nD}$ denote the collection of filters for each value of $n$, where $D = 300$ is the word vector dimensionality. If $\mathbf{v}_i^n$ denotes the concatenation of $n$ word vectors starting at index $i$, let $\mathbf{m}_n = [\mathbf{v}_1^n; \mathbf{v}_2^n; \ldots; \mathbf{v}_{k_u - n + 1}^n]$ be the list of $n$-grams that convolutional filters of length $n$ run over. The three intermediate representations are then given by:

$$R_n = F_n^s \, \mathbf{m}_n \qquad (5)$$

Each column of the intermediate matrices $R_n$ is produced by a single convolutional filter of length $n$. We obtain summary $n$-gram representations by pushing these representations through a rectified linear unit (ReLu) activation function (Nair and Hinton, 2010) and max-pooling over time (i.e. columns of the matrix) to get a single feature for each of the $L$ filters applied to the utterance:

$$\mathbf{r}_n' = \texttt{maxpool}\left(\texttt{ReLu}\left(R_n + b_n^s\right)\right) \qquad (6)$$

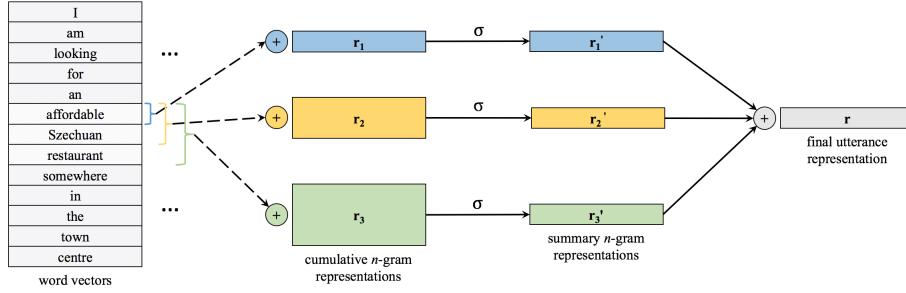

Figure 3: NBT-DNN Model. Word vectors of all unigrams, bigrams and trigrams are summed to obtain *cumulative* $n$-gram representations. These are passed through another hidden layer and then summed to obtain the utterance representation $\mathbf{r}$.

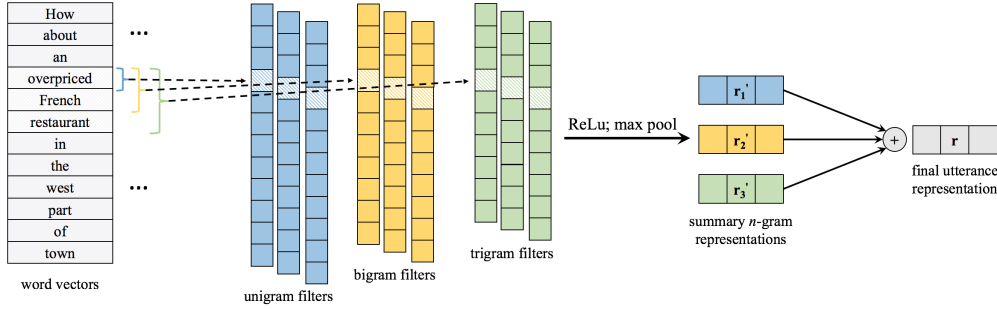

Figure 4: NBT-CNN Model. $L$ convolutional filters of window sizes $1, 2, 3$ are applied to word vectors of the given utterance ($L = 3$ in the diagram, but $L = 300$ in the system). The convolutions are followed by the ReLu activation function and max-pooling to produce summary $n$-gram representations. These are summed to obtain the utterance representation $\mathbf{r}$.

where $b_n^s$ is a bias term broadcast across all filters. Finally, the three summary $n$-gram representations are summed to obtain the final utterance representation vector $\mathbf{r}$ (as in Equation 4). The NBT-CNN model is (by design) better suited to longer utterances, as its convolutional filters interact directly with subsequences of the utterance, and not just their noisy summaries given by the NBT-DNN's cumulative $n$-grams.

## 3.2 Semantic Decoding

The NBT diagram in Figure 2 shows that the utterance representation $\mathbf{r}$ and the candidate slot-value pair representation $\mathbf{c}$ directly interact through the *semantic decoding* module. This component decides whether the user explicitly expressed an intent matching the current candidate pair (i.e. without taking the dialogue context into account). Examples of such matches would be '*I want Thai food*' with `food=Thai` or more demanding ones such as '*a pricey restaurant*' with `price=expensive`. This is where the use of high-quality pre-trained word vectors comes into play: a delexicalisation-based model could deal with the former example but would be helpless in the latter case, unless a human expert had provided a semantic dictionary listing all potential rephrasings for each value in the domain ontology.

Let the vector space representations of a candidate pair's slot name and value be given by $\mathbf{c_s}$ and $\mathbf{c_v}$ (with vectors of multi-word slot names/values summed together). The NBT model learns to map this tuple into a single vector $\mathbf{c}$ of the same dimensionality as the utterance representation $\mathbf{r}$. These two representations are then forced to interact in order to learn a similarity metric which discriminates between interactions of utterances with slot-value pairs that they either do or do not express:

$$\mathbf{c} = \sigma \left( W_c^s(\mathbf{c_s} + \mathbf{c_v}) + b_c^s \right) \qquad (7)$$

$$\mathbf{d} = \mathbf{r} \otimes \mathbf{c} \qquad (8)$$

where $\otimes$ denotes *element-wise* vector multiplication. The dot product, which may seem like the more intuitive similarity metric, would reduce the rich set of features in $\mathbf{d}$ to a single scalar. The element-wise multiplication allows the downstream network to make better use of its parameters by learning non-linear interactions between sets of features in $\mathbf{r}$ and $\mathbf{c}$.[2] The downstream network (Binary Decision Maker in Figure 2) uses one intermediate hidden layer of size 100 to make the final binary decision regarding the current candidate pair.

---

[2] We also tried to concatenate $\mathbf{r}$ and $\mathbf{c}$ and pass that vector to the downstream decision-making neural network. However, this set-up led to very weak performance since our relatively small datasets did not suffice for the network to learn to model the interaction between the two feature vectors.

### 3.3 Context Modelling

This 'decoder' does not yet suffice to extract intents from utterances in human-machine dialogue. To understand some queries, the belief tracker must be aware of *context*, i.e. the flow of dialogue leading up to the latest user utterance. While all previous system and user utterances are important, the most relevant one is the last system utterance, in which the dialogue system could have performed (among others) one of the following two *system acts*:

1. **System Request**: The system asks the user about the value of a specific slot $T_q$. If the system utterance is: *'what price range would you like?'* and the user answers with *any*, the model must infer the reference to *price range*, and not to other slots such as *area* or *food*.

2. **System Confirm:** The system asks the user to confirm whether a specific slot-value pair $(T_s, T_v)$ is part of their desired constraints. For example, if the user responds to *'how about Turkish food?'* with *'yes'*, the model must be aware of the system act in order to correctly update the belief state.

If we make the Markovian decision to only consider the last set of system acts, we can incorporate context modelling into the NBT model. Let $\mathbf{t_q}$ and $(\mathbf{t_s}, \mathbf{t_v})$ be the word vector representations of the arguments for the system request and confirm acts (zero vectors if none). The model computes the following measures of similarity between the system acts, candidate pair $(\mathbf{c_s}, \mathbf{c_v})$ and utterance representation $\mathbf{r}$:

$$\mathbf{d_r} = (\mathbf{c_s} \cdot \mathbf{t_q})\mathbf{r} \qquad (9)$$
$$\mathbf{d_c} = (\mathbf{c_s} \cdot \mathbf{t_s})(\mathbf{c_v} \cdot \mathbf{t_v})\mathbf{r} \qquad (10)$$

where $\cdot$ denotes dot product. The computed similarity terms act as gating mechanisms which only pass the utterance representation through if the system asked about the current candidate slot or slot-value pair. This type of interaction is particularly useful for the confirm system act: if the system asks the user to confirm, the user is likely not to mention any slot values, but to just respond affirmatively or negatively. This means that the model must consider the *three-way interaction* between the utterance, candidate slot-value pair and the slot value pair offered by the system. If (and only if) the latter two are the same should the model consider the affirmative or negative polarity of the user utterance when

making the subsequent binary decision. Finally, these two context modelling summary representations are passed to the decision making module, which combines them with the semantic decoder output $\mathbf{d}$ to make the final binary decision.

## 4 Belief State Update Mechanism

In spoken dialogue systems, belief tracking models operate over the output of automatic speech recognition (ASR). Despite improvements to speech recognition, the need to make the most out of imperfect ASR will persist as dialogue systems are used in increasingly noisy environments.

In this work, we define a simple rule-based belief state update mechanism which can be applied to ASR $N$-best lists. For dialogue turn $t$, let $sys^{t-1}$ denote the preceding system output, and let $h^t$ denote the list of $N$ ASR hypotheses $h_i^t$ with posterior probabilities $p_i^t$. For any hypothesis $h_i^t$, slot $s$ and slot value $v \in V_s$, NBT models estimate $\mathbb{P}(s, v \mid h_i^t, sys^{t-1})$, which is the (turn-level) probability that $(s, v)$ was expressed in the given hypothesis. The predictions for $N$ such hypotheses are then combined as:

$$\mathbb{P}(s, v \mid h^t, sys^{t-1}) = \sum_{i=1}^{N} p_i^t \, \mathbb{P}\left(s, v \mid h_i^t, sys^t\right)$$

This turn-level belief state estimate is then combined with the (cumulative) belief state up to time $(t-1)$ to get the updated belief state estimate:

$$\mathbb{P}(s, v \mid h^{1:t}, sys^{1:t-1}) = \lambda \, \mathbb{P}\left(s, v \mid h^t, sys^{t-1}\right) \\ + (1-\lambda) \, \mathbb{P}\left(s, v \mid h^{1:t-1}, sys^{1:t-2}\right)$$

where $\lambda$ is the coefficient which determines the relative weight of the turn-level and previous turns' belief state estimates.[3] For slot $s$, the set of its *detected values* at turn $t$ is then given by:

$$V_s^t = \{v \in V_s \mid \mathbb{P}\left(s, v \mid h^{1:t}, sys^{1:t-1}\right) \geq 0.5\}$$

For informable (i.e. goal-tracking) slots, the value in $V_s^t$ with the highest probability is chosen as the current goal (if $V_s^t \neq \{\emptyset\}$). For requests, all slots in $V_{req}^t$ are deemed to have been requested. As requestable slots serve to model single-turn user queries, they require no belief tracking across turns.

---

[3]This coefficient was tuned on the DSTC2 development set. The best performance was achieved with $\lambda = 0.55$.

# 5 Experiments

## 5.1 Datasets

Two datasets were used for training and evaluation. Both consist of user conversations with task-oriented dialogue systems designed to help users find suitable restaurants around Cambridge, UK. The two corpora share the same domain ontology, which contains three *informable* (i.e. goal-tracking) slots: FOOD TYPE, AREA and PRICE. The users can specify values for these slots in order to find restaurants which best meet their criteria. Once the system suggests a restaurant, the users can ask about the values of up to eight *requestable* slots (PHONE NUMBER, ADDRESS, etc.). The two datasets are:

**1. DSTC2**: We use the transcriptions, ASR hypotheses and turn-level semantic labels provided for the Dialogue State Tracking Challenge 2 (Henderson et al., 2014a). The official transcriptions contain various spelling errors which we corrected manually; the cleaned version of the dataset will be made available on the first author's website. The training data contains 2207 dialogues and the test set consists of 1117 dialogues. We train NBT models on transcriptions and report belief tracking performance on test set ASR predictions.

**2. WOZ 2.0**: Wen et al. (2017) performed a Wizard of Oz style experiment in which Amazon Mechanical Turk users assumed the role of the system or the user of a task-oriented dialogue system based on the DSTC2 ontology. Users typed instead of using speech, which means performance in the WOZ experiments is more indicative of the model's capacity for semantic understanding than its robustness to ASR errors. Whereas in the DSTC2 dialogues users would quickly adapt to the system's (lack of) language understanding capability, the WOZ experimental design gave them freedom to use more sophisticated language. We expanded the original WOZ dataset from Wen et al. (2017) using the same data collection procedure, yielding a total of 1200 dialogues (5012 turns). We divided these into 600 training, 200 validation and 400 test set dialogues. The WOZ 2.0 dataset will also be available from the first author's website.

We focus on two key evaluation metrics introduced in (Henderson et al., 2014a):

1. **Goals** ('joint goal accuracy'): the proportion of dialogue turns where all the user's search goal constraints were correctly identified;

2. **Requests**: similarly, the proportion of dialogue turns where user's requests for information were identified correctly.

## 5.2 Models

We evaluate two NBT model variants: NBT-DNN and NBT-CNN. The two corpora are used to create training data for two separate experiments. We iterate over all utterances, generating one example for each of the slot-value pairs in the ontology. An example consists of a transcription, its context (i.e. list of preceding system acts) and a candidate slot-value pair. The binary label for each example indicates whether or not its utterance and context express the example's candidate pair. To train the models, we used the Adam optimizer (Kingma and Ba, 2015) with cross-entropy loss, backpropagating through all the NBT subcomponents while keeping the pre-trained word vectors fixed.

For each dataset, we compare the NBT models to: **1)** a *baseline* system that implements a well-known competitive delexicalisation-based model for that dataset; and **2)** the same baseline model supplemented with a task-specific semantic dictionary (produced by the baseline system creators). For DSTC2, the model is that of Henderson et al. (2014d; 2014c). This model is an $n$-gram based neural network model with recurrent connections between turns (but not inside utterances) which replaces occurrences of slot names and values with generic delexicalised features. For WOZ 2.0, we compare the NBT models to a more sophisticated belief tracking model presented in (Wen et al., 2017). This model uses an RNN for belief state updates and a CNN for turn-level feature extraction. Unlike NBT-CNN, their CNN operates not over vectors, but over delexicalised features akin to those used by Henderson et al. (2014c).

Both baseline models map exact matches of ontology-defined intents (and their lexicon-specified rephrasings) to one-hot delexicalised $n$-gram features. This means that pre-trained vectors cannot be incorporated directly into these models.

# 6 Results

## 6.1 Belief Tracking Performance

Table 1 shows the performance of NBT models trained and evaluated on DSTC2 and WOZ 2.0 datasets. The NBT models outperformed the baseline models in terms of both joint goal and request accuracies. For goals, the gains are *always* statis-

| Model | DSTC2 | | WOZ 2.0 | |
|---|---|---|---|---|
| | Goals | Requests | Goals | Requests |
| **Baseline DST** | 69.1 | 95.7 | 70.8 | 87.1 |
| **+ sem. dict.** | 72.9* | 95.7 | 83.7* | 87.6 |
| **NBT-DNN** | 72.6* | 96.4 | **84.4*** | 91.2* |
| **NBT-CNN** | 73.4* | 96.5 | 84.2* | **91.6*** |

Table 1: DSTC2 and WOZ 2.0 test set accuracies for: **a)** joint goals; and **b)** turn-level requests. The asterisk indicates statistically significant improvement over the baseline delexicalisation-based trackers (paired $t$-test; $p < 0.05$).

| Word Vectors | DSTC2 | | WOZ 2.0 | |
|---|---|---|---|---|
| | Goals | Requests | Goals | Requests |
| XAVIER | 64.2 | 81.2 | 81.2 | 90.7 |
| GloVe | 69.0* | 96.4* | 80.1 | 91.4 |
| Paragram-SL999 | **73.4*** | **96.5*** | **84.2*** | 91.6 |

Table 2: DSTC2 and WOZ 2.0 test set performance (*joint goals* and *requests*) of the NBT-CNN model making use of three different word vector collections. The asterisk indicates statistically significant improvement over the baseline XAVIER (random) word vectors (paired $t$-test; $p < 0.05$).

tically significant (paired $t$-test, $p < 0.05$). Moreover, there was no statistically significant variation between the NBT and the lexicon-supplemented models, showing that the NBT can handle semantic relations which otherwise had to be explicitly encoded in semantic dictionaries.

While the NBT performs well across the board, we can compare its performance on the two datasets to understand its strengths. The improvement over the baseline is greater on WOZ 2.0, which corroborates our intuition that the NBT's ability to learn linguistic variation is vital for this dataset containing longer sentences, richer vocabulary and no ASR errors. By comparison, the language of the subjects in the DSTC2 dataset is less rich, and compensating for ASR errors is the main hurdle: given access to the DSTC2 test set transcriptions, the NBT models' goal accuracy rises to 0.96. This indicates that future work should focus on better ASR compensation if the model is to be deployed in environments with challenging acoustics.

### 6.2 The Importance of Word Vector Spaces

The NBT models use the semantic relations embedded in the pre-trained word vectors to handle semantic variation and produce high-quality intermediate representations. Table 2 shows the performance of NBT-CNN[4] models making use of three different word vector collections: **1)** 'random' word vectors initialised using the XAVIER initialisation (Glorot and Bengio, 2010); **2)** distributional GloVe vectors (Pennington et al., 2014), trained using co-occurrence information in large textual corpora; and **3)** *semantically specialised* Paragram-SL999 vectors (Wieting et al., 2015), which are obtained by injecting *semantic similarity constraints* from the Paraphrase Database (Ganitkevitch et al., 2013) into distributional word vectors in order to improve their semantic content.

The results in Table 2 show that the use of seman-

---

[4]The NBT-DNN model showed the same trends. For brevity, Table 2 presents only the NBT-CNN figures.

tically specialised word vectors leads to considerable performance gains: Paragram-SL999 vectors (significantly) outperformed GloVe and XAVIER vectors for goal tracking on both datasets. The gains are particularly robust for noisy DSTC2 data, where both collections of pre-trained vectors consistently outperformed random initialisation. The gains are weaker for the noise-free WOZ 2.0 dataset, which seems to be large (and clean) enough for the NBT model to learn task-specific rephrasings and compensate for the lack of semantic content in the word vectors. For this dataset, GloVe vectors do not improve over the randomly initialised ones. We believe this happens because distributional models keep related, yet antonymous words close together (e.g. *north* and *south*, *expensive* and *inexpensive*), offsetting the useful semantic content embedded in this vector spaces.

### 7 Conclusion

In this paper, we have proposed a novel neural belief tracking (NBT) framework designed to overcome current obstacles to deploying dialogue systems in real-world dialogue domains. The NBT models offer the known advantages of coupling Spoken Language Understanding and Dialogue State Tracking, without relying on hand-crafted semantic lexicons to achieve state-of-the-art performance. Our evaluation demonstrated these benefits: the NBT models match the performance of models which make use of such lexicons and vastly outperform them when these are not available. Finally, we have shown that the performance of NBT models improves with the semantic quality of the underlying word vectors. To the best of our knowledge, we are the first to move past intrinsic evaluation and show that *semantic specialisation* boosts performance in downstream tasks.

In future work, we intend to explore applications of the NBT for multi-domain dialogue systems, as well as in languages other than English that require handling of complex morphological variation.

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
