# Peer review of "Neural Belief Tracker: Data-Driven Dialogue State Tracking"

_ACL 2017 — decision unknown_

[Official Review · Reviewer 1 · rating 4 · confidence 4]
soundness 4 · originality 3 · clarity 4 · impact 3 · substance 4 · appropriateness 5 · meaningful comparison 2 · presentation format Oral Presentation

- Strengths:
This paper proposes a novel approach for dialogue state tracking that benefits
from representing slot values with pre-trained embeddings and learns to compose
them into distributed representations of user utterances and dialogue context.
Experiments performed on two datasets show consistent and significant
improvements over the baseline of previous delexicalization based approach.
Alternative approaches (i.e., XAVIER, GloVe, Program-SL999) for pre-training
word embeddings have been investigated.

- Weaknesses:
Although one of the main motivations for using embeddings is to generalize to
more complex dialogue domains where delexicalization may not scale for, the
datasets used seem limited.    I wonder how the approach would compare with and
without a separate slot tagging component on more complex dialogues. For
example, when computing similarity between the utterance and slot value pairs,
one can actually limit the estimation to the span of the slot values. This
should be applicable even when the values do not match.

I think the examples in the intro is misleading, shouldn’t the dialogue state
also include “restaurant_name=The House”? This brings another question, how
does resolution of coreferences impact this task?

- General Discussion:
On the overall, use of pre-trained word embeddings is a great idea, and the
specific approach for using them is exciting.

[Official Review · Reviewer 2 · rating 4 · confidence 5]
soundness 4 · originality 3 · clarity 3 · impact 3 · substance 3 · appropriateness 5 · meaningful comparison 2 · presentation format Poster

This paper presents a neural network-based framework for dialogue state
tracking.
The main contribution of this work is on learning representations of user
utterances, system outputs, and also ontology entries, all of which are based
on pre-trained word vectors.
Particularly for the utterance representation, the authors compared two
different neural network models: NBT-DNN and NBT-CNN.
The learned representations are combined with each other and finally used in
the downstream network to make binary decision for a given slot value pair.
The experiment shows that the proposed framework achieved significant
performance improvements compared to the baseline with the delexicalized
approach.

It's generally a quality work with clear goal, reasonable idea, and improved
results from previous studies.
But the paper itself doesn't seem to be very well organized to effectively
deliver the details especially to readers who are not familiar with this area.

First of all, more formal definition of DST needs to be given at the beginning
of this paper.
It is not clear enough and could be more confusing after coupling with SLU.
My suggestion is to provide a general architecture of dialogue system described
in Section 1 rather than Section 2, followed by the problem definition of DST
focusing on its relationships to other components including ASR, SLU, and
policy learning.

And it would also help to improve the readability if all the notations used
throughout the paper are defined in an earlier section.
Some symbols (e.g. t_q, t_s, t_v) are used much earlier than their
descriptions.

Below are other comments or questions:

- Would it be possible to perform the separate SLU with this model? If no, the
term 'joint' could be misleading that this model is able to handle both tasks.

- Could you please provide some statistics about how many errors were corrected
from the original DSTC2 dataset?
If it is not very huge, the experiment could include the comparisons also with
other published work including DSTC2 entries using the same dataset.

- What do you think about using RNNs or LSTMs to learn the sequential aspects
in learning utterance representations?
Considering the recent successes of these recurrent networks in SLU problems,
it could be effective to DST as well.

- Some more details about the semantic dictionary used with the baseline would
help to imply the cost for building this kind of resources manually.

- It would be great if you could give some samples which were not correctly
predicted by the baseline but solved with your proposed models.